# Study of Albumin Oxidation in COVID-19 Pneumonia Patients: Possible Mechanisms and Consequences

**DOI:** 10.3390/ijms231710103

**Published:** 2022-09-03

**Authors:** Tomasz Wybranowski, Marta Napiórkowska, Maciej Bosek, Jerzy Pyskir, Blanka Ziomkowska, Michał Cyrankiewicz, Małgorzata Pyskir, Marta Pilaczyńska-Cemel, Milena Rogańska, Stefan Kruszewski, Grzegorz Przybylski

**Affiliations:** 1Department of Biophysics, Faculty of Pharmacy, Collegium Medicum in Bydgoszcz, Nicolaus Copernicus University in Toruń, 85-067 Bydgoszcz, Poland; 2Department of Rehabilitation, Faculty of Health Sciences, Collegium Medicum in Bydgoszcz, Nicolaus Copernicus University in Toruń, 85-094 Bydgoszcz, Poland; 3Department of Lung Diseases, Neoplasms and Tuberculosis, Faculty of Medicine, Collegium Medicum in Bydgoszcz, Nicolaus Copernicus University in Toruń, 85-326 Bydgoszcz, Poland

**Keywords:** COVID-19, oxidative stress, albumin, advanced oxidation protein products, advanced lipoxidation end-products, chloramine T, malondialdehyde

## Abstract

Oxidative stress induced by neutrophils and hypoxia in COVID-19 pneumonia leads to albumin modification. This may result in elevated levels of advanced oxidation protein products (AOPPs) and advanced lipoxidation end-products (ALEs) that trigger oxidative bursts of neutrophils and thus participate in cytokine storms, accelerating endothelial lung cell injury, leading to respiratory distress. In this study, sixty-six hospitalized COVID-19 patients with respiratory symptoms were studied. AOPPs-HSA was produced in vitro by treating human serum albumin (HSA) with chloramine T. The interaction of malondialdehyde with HSA was studied using time-resolved fluorescence spectroscopy. The findings revealed a significantly elevated level of AOPPs in COVID-19 pneumonia patients on admission to the hospital and one week later as long as they were in the acute phase of infection when compared with values recorded for the same patients 6- and 12-months post-infection. Significant negative correlations of albumin and positive correlations of AOPPs with, e.g., procalcitonin, D-dimers, lactate dehydrogenase, aspartate transaminase, and radiological scores of computed tomography (HRCT), were observed. The AOPPs/albumin ratio was found to be strongly correlated with D-dimers. We suggest that oxidized albumin could be involved in COVID-19 pathophysiology. Some possible clinical consequences of the modification of albumin are also discussed.

## 1. Introduction

Coronavirus disease 2019 (COVID-19), caused by severe acute respiratory syndrome coronavirus 2 (SARS-CoV-2), was proclaimed a critical global pandemic by the World Health Organization (WHO) in March 2020. Increasingly, recent evidence indicates the presence of higher levels of inflammatory cytokines in COVID-19 patients with critical and severe disease than in moderately ill patients and healthy individuals. This “cytokine storm” can also indicate a poor prognosis and may increase the mortality rate of COVID-19 patients [1,2,3,4]. The entry of SARS-CoV-2 into cells may result in the development of a cytokine storm in the host body, characterized by a high plasma level of pro-inflammatory cytokines, including interleukin (IL)-6, IL-2, IL-7, IL-10, monocyte chemoattractant protein-1 (MCP-1), macrophage inflammatory protein-1A (MIP-1A), tumor necrosis factor-alpha (TNF-alfa), and interferon-gamma inducible protein (IP10) [5,6]. A bronchoalveolar lavage taken from patients with severe COVID-19-related pneumonia showed a high level of chemokines being secreted from macrophages [7]. Post-mortem analysis of lung tissue of patients with severe COVID-19-related pneumonia also found an excessive amount of immune cell infiltration [8]. Respiratory viral infections are, in general, associated with proinflammatory cytokine production, inflammation, cell apoptosis, and other pathophysiological processes, which lead to high levels of oxidative stress [9]. Indeed, some studies have revealed that SARS-CoV-2 infection pathogenesis is related to oxidative stress [10,11,12]. Furthermore, COVID-19 patients with severe pulmonary involvement showed a higher level of endogenous oxidative markers than patients with mild disease [13]. There is a strong link between inflammatory state, activated neutrophils, oxidative stress, oxidized albumin, and decreased albumin concentration [10]. It is still premature to confirm whether or not lung involvement or signs of ongoing heart inflammation occur as a temporary response to COVID-19 infection and will spontaneously be resolved over time and whether COVID-19 affects other inflammation-related complications or basic diseases, such as intestinal diseases [14,15,16,17]. Oxidative stress may also play an important role in the development of symptoms from Long COVID [18].

Albumin serves as a major anti-inflammatory agent in our bodies [19,20], and one of its properties that has been less discussed in the literature is its anti-oxidative and anti-thrombotic activity [21,22]. It may also exhibit antiviral properties, as recent studies have revealed that albumin specifically binds the SARS-CoV-2 spike protein S1 subunit [23]. Human serum albumin (HSA) comprises the largest thiol pool in plasma and thus determines the plasma redox status [24,25]. It was determined that oxidized albumin is the direct mediator of triggering inflammation, leading to neutrophil activation and thus probably increasing oxidative stress [26,27,28,29]. The pro-inflammatory properties of in vivo-oxidized albumin initiating vascular injury were also examined [27]. Furthermore, recently, more researchers have claimed that oxidized albumin is more than an oxidative stress biomarker and can be an aggravating factor for various diseases [30]. Recently, there has been a paradigm shift regarding our knowledge of albumin and its role in organisms. New observations have revealed that the form of albumin is also of great importance in the pathogenesis of diseases [31]. It has also been observed that oxidized albumin promotes inflammation by the modulation of platelets [32,33]. 

Several researchers suggested at the beginning of the COVID-19 pandemic that oxidized albumin may be “an opportunity for diagnoses or treatment of COVID-19” and “a positive predictor of mortality” [30]. Thus far, there have been few reports on the oxidation of albumin in COVID-19 infection [34,35]. A very good association of decreased thiol concentrations and increased amounts of advanced oxidation protein products (AOPPs) with COVID-19 severity, intensive care unit admission criteria, and mortality has been revealed [36]. Furthermore, the majority of investigations have only focused on the effects of hypoalbuminemia on mortality and prognosis of COVID-19 severity [37]. It has been observed that decreased albumin concentration may be a risk factor for mortality [37,38,39,40] and may be related to inflammation in several diseases [41,42,43]. A negative relationship between serum albumin and C-reactive protein (CRP), which is an acute-phase reactant produced in response to inflammation state and pro-inflammatory IL-6 or TNF-alpha, is often identified in patients hospitalized for many reasons [44,45,46,47]. However, the mechanism of hypoalbuminemia in COVID-19 has not yet been fully explained [38,48]. It should be noted that hypoalbuminemia may not only be associated with abnormal liver function and diminished albumin synthesis in inflammation but may also be a consequence of a high clearance of damaged and oxidized albumin [31,49]. One pharmacokinetic analysis showed that albumin oxidized by chloramine T left the circulation very quickly after intravenous injection and accumulated mainly in the liver, kidneys, and spleen [50].

AOPPs have been found to be a marker of the intensity of inflammation, used to predict the course of the disease, determined to act as a mediator of the activation of monocytes, and hypothesized to stimulate a respiratory burst of neutrophils [51,52,53,54]. AOPPs have also been reported to be strongly correlated with levels of neopterin, which is produced by macrophages upon stimulation with interferon-γ [51].

We hypothesized that oxidized albumin could be involved in the pathophysiology of COVID-19 infection. The aim of the study was to assess AOPPs levels in COVID-19 pneumonia patients on admission to the hospital and their correlations with inflammatory biomarkers, radiological scores of computed tomography (HRCT), multiorgan impairment biomarkers, and other surrogate markers of severity or mortality in COVID-19. Additionally, the level of AOPPs on admission was compared with values recorded for the same patients one week later, as long as they were in the acute phase of infection and 6 and 12 months post-infection. To demonstrate the mechanism of formation of AOPPs in vivo, in our study, HSA was incubated with increasing concentrations of chloramine T. The structure of AOPPs is similar to that obtained by artificial HSA oxidation by chloramine T [50,51,55]. The chlorine compounds produced by neutrophils in inflammation are directly associated with albumin oxidative modifications and the creation of AOPPs [29,55].

Another important modification of HSA that may alter albumin function and influence the severity of symptoms in patients with COVID-19-related pneumonia is related to the creation of MDA-HSA adducts, known as advanced lipoxidation end-products (ALEs). In our study, ALEs were produced in vitro by incubating HSA with malondialdehyde (MDA). Malondialdehyde (MDA) is generated during secondary lipid oxidation. ALEs-HSA can induce monocyte activation and vascular complications due to its pro-inflammatory effect [56,57]. Advanced glycation end products (AGEs) as well as ALEs and AOPPs can bind to a specific receptor called RAGE, which causes the upregulation of inflammatory pathways [57,58,59,60]. This phenomenon has barely been examined so far. ALEs-HSA may also be involved in increasing the production of autoantibodies in some autoimmune disorders, such as systemic lupus erythematosus and arthritis [61,62]. It is well known that aldehydes are capable of binding to proteins and forming fluorescent compounds with a maximum emission wavelength of 440–480 nm [63,64]. Time-resolved fluorescence spectroscopy has been used to study the interaction between MDA and HSA (ALEs-HSA formation). This time-resolved measurement can reveal fluorescence intensity decay in terms of lifetimes. The fluorescence lifetime is sensitive to the local environment of the fluorophore and may vary, for example, due to conformational changes in molecules and during molecular interactions with other molecules [65]. To the best of our knowledge, there are no reports of the use of time-resolved spectroscopy in the study of the autofluorescence of MDA-modified proteins.

## 2. Results

### 2.1. Clinical Study

Our results revealed that the values of most clinical parameters typically used in predicting the mortality and severity in COVID-19 patients were significantly higher in patients with COVID-19-related pneumonia compared to reference values. As shown in Table 1, the level of albumin was found to be negatively correlated with the clinical biomarkers procalcitonin (PCT), D-dimers, lactate dehydrogenase (LDH), troponin, and aspartate transaminase (AST). The results indicate that there was also a strong negative association between albumin and the radiological score of computed tomography (HRCT). The AOPPs measured in COVID-19 patients upon hospital admission were found to be positively correlated with liver enzymes (AST, alanine transaminase (ALT)), D-dimers, LDH, CRP, PCT, and HRCT score.

During the week of hospitalization, the AOPPs levels increase significantly (*p* < 0.001) (Figure 1, Table 2). Six months after COVID-19 infection, the AOPPs levels in the same patient group decreased significantly (*p* < 0.001). A slight increase, though not statistically significant, was observed at the next 6-month follow-up (*p* = 0.055).

### 2.2. In Vitro Study

To investigate the impact of oxidative stress and carbonyl stress on the properties of HSA, several experiments were performed. In our study, AOPPs-HSA were produced in vitro by treating HSA with oxidants (chloramine T).

It can be seen in Figure 2 that the addition of chloramine T to HSA resulted in the formation of AOPPs. In this figure, one can also observe the exponential nature of the relationship between the level of AOPPs and the concentration of chloramine T.

Figure 3a shows the normalized fluorescence decay curves of ALEs-HSA at 450 nm for different lengths of incubation after the addition of MDA to HSA, which was carried out at 37 °C. The decay curves were fitted to a three-exponential fluorescence model (χ^2^ = 1). The estimated mean fluorescence lifetime is presented in Figure 3b. The changes were observed over 15 h of incubation before equilibrium was established.

## 3. Discussion

On the basis of the results presented in Figure 1 and Table 1, it can be unequivocally concluded that elevated levels of AOPPs are associated with inflammation during the course of COVID-19-related pneumonia. Six and twelve months after infection, significantly decreased levels of AOPPs were observed when compared to the COVID-19 patients during the acute infection phase. It can be ruled out that the AOPPs levels long after the diagnosis were lower than prior to the infection due to long-term chronic symptoms following COVID-19. The role of oxidative stress in the pathophysiology of Long COVID has not been yet elucidated, and further research is needed. The level of AOPPs is raised in many diseases, including renal failure, diabetes, and atherosclerosis, but also increases with age [66,67,68]. Very often, a severe course of COVID-19 is associated with comorbidities [69,70]. Therefore, an elevated AOPPs level prior to infection may also be a risk factor for severe COVID-19. Interestingly, the level of AOPPs in this study showed an increasing dependence on time from the onset of symptoms prior to the admission of patients to their arrival in the hospital department (r = 0.318). It is suggested that activated neutrophils in conditions of infection and inflammation are the main cause of the enhanced production of free radicals (e.g., hypochlorous acid (HOCl)) and increased oxidative stress observed in critically ill COVID-19 patients. The accumulation of reactive oxygen species (ROS) within neutrophils is considered a key process in the initiation of neutrophil extracellular traps (NETs), which are able to entrap a wide variety of pathogens and prevent their dissemination into the blood circulation [71]. In one study, sera from patients with COVID-19 triggered the release of NETs from control neutrophils in vitro. Another interesting finding was that the oxidation of albumin also activated the release of NETs from control neutrophils [72]. In a previous study, neutrophil activation and the formation of NETs were reported as major risk factors for acute lung injury [73]. Early CT findings from COVID-19 patients may correspond to viscous secretions seeping through the pulmonary alveoli due to hyperinflammation in the lung and the overproduction of NETs [74]. Interestingly, we found a weak but significant positive correlation between the level of AOPPs and HRCT scores (r = 0.348).

The results of our in vitro studies shown in Figure 2 indicate that chloramine T produces AOPPs-HSA in an exponential dose-dependent manner. The shape of this curve may explain the rapid course of the “cytokine storm” observed in COVID-19 patients. This experiment provides insight into the mechanism underlying severe COVID-19 pathology. HSA modified by chlorine compounds is considered a pro-inflammatory mediator due to its induction of a neutrophil respiratory burst [29]. Chlorine-induced albumin damage appears to play a key role in exacerbating respiratory problems in COVID-19 patients. It should also be emphasized that the contribution of individual AOPP products is probably not the same for different concentrations of chloramine T, and that dityrosine crosslinks are formed only at high concentrations of chloramine T in relation to HSA.

We hypothesize that the production of ALEs-HSA may also contribute to the aggravation of inflammation in COVID-19, as some pro-inflammatory effects of ALEs have been identified [56,57,58,59,60,75]. ROS are produced in cells more rapidly due to their viral entry into the cytoplasm of the host epithelial cells. ROS production might be activated either by viral components or by cytokines [12]. Oxidative stress in cells leads to lipid peroxidation, which in turn produces aldehyde products. In some studies, higher lipid peroxidation levels and increases in the concentration of aldehydes were found to be associated with an increased severity of disease in COVID-19 patients [76]. Furthermore, protein adducts of the lipid peroxidation product 4-hydroxynonenal (HNE) were found to be present at higher volumes in the plasma of COVID-19 patients who did not survive [77]. Albumin has the ability to conjugate with aldehydes to form Schiff bases or Michael adducts by generating covalent adducts [78]. Aldehydes can react with protein and the sulfhydryl group of cysteine, thiols, and amine groups in both oxidative (lipoxidation) and non-oxidative reactions [78,79,80,81]. Furthermore, the addition of toxic lipid peroxidation by-product (e.g., MDA) to proteins results in an increase in carbonyl contents [82]. To better understand the formation of ALEs-HSA, time-resolved fluorescence spectroscopy was used in our study. MDA modifies HSA and generates fluorescent products exhibiting a specific fluorescence emission wavelength between 400 and 550 nm when the sample is excited at 360 nm. The results shown in Figure 2 reveal that the stable ALEs-HSA were not formed immediately after the addition of MDA to the HSA solution but instead after about 15 h of incubation. These results suggest that the interaction between MDA and HSA involves more than the simple formation of covalent MDA-HSA adducts and is likely to occur in several steps, giving rise to more complex ALEs-HSA. It is likely that many products with unknown molecular structures are created.

The degree of inflammation caused by virus entry into cells is linked to the overproduction of poly(ADP-ribose) polymerase 1 (PARP-1), which is a key regulator of the virus life cycle. The PARP-1-mediated synthesis of ADP-ribose chains reduces the level of nicotinamide adenine dinucleotide (NAD+) [83,84]. The alteration of NAD+ may impair the detoxification of aldehydes by dehydrogenases, as NAD+ is a redox cofactor for these enzymes. This can lead to an increase in the modification of albumin by aldehydes in infected cells. It should be noted that the NAD+ level also declines with age due to inflammatory responses related to senescent cells, which produce a permanent alarm signal [85,86,87]. Many comorbidities are also associated with chronic inflammation, which causes a reduction in the level of NAD+ [88]. Some researchers directly suggest that NAD+ deficiency may be a major mortality risk factor in COVID-19 patients [83,84,89]. As NAD+ is considered to be a regulator of immune responses during viral infections, its use as a drug in the treatment of COVID-19 has been proposed [83,90].

Shortness of breath progressing to hypoxic respiratory failure is a common clinical manifestation observed in patients with pneumonia. A fall in oxygen saturation (hypoxemia) and higher respiratory rate (tachypnea) are associated with increased mortality risk in COVID-19 patients [91,92]. Hypoxia can occur even without dyspnea, and this phenomenon is called silent hypoxia [93,94]. Hypoxic cells produce larger amounts of aldehydes and ROS, leading to an even greater molecular modification of albumin [95]. Interestingly, some studies have revealed connections between AOPPs and hypoxia. Higher amounts of AOPPs have also been found in hypoxic newborn infants than in controls [96].

In the case of damage to the liver and other organs involved in the removal of modified albumin, high amounts of AOPPs-HSA and ALEs-HSA may not be cleared from the bloodstream as quickly as they would be in a healthy person. Liver function test abnormalities are very common in COVID-19-related pneumonia patients [92,97,98]. This may be related to liver injury by the virus itself, inflammatory responses, hepatic ischemia, hepatic hypoxia, or even tissue damage and muscle breakdown. In particular, an elevated activity of LDH reflects tissue destruction and is regarded as a prognostic marker of outcomes in COVID-19 patients [92,99]. Regarding our biochemical parameters, LDH levels were also found to be markedly above normal, indicating tissue injury. A significant correlation of the liver enzyme AST with LDH (r = 0.395) was found. AST was also found to be moderately negatively associated with albumin (r = −0.307) and positively associated with AOPPs (r = 0.362). In conditions of both inflammation and liver damage, the synthesis of albumin may also be disturbed, as albumin is produced exclusively by liver cells.

It should also be noted that high-molecular-weight AOPPs can influence the aggregation of red blood cells (RBCs) and the formation of blood clots. High-molecular-weight AOPPs are formed due to the tendency of albumin to form aggregates via disulfide bridges and/or dityrosine cross-linking [52]. An in vitro study showed that chlorine active species induce the formation of high-molecular-weight proteins [55]. The modification of HSA with MDA can also result in the aggregation of albumin [64]. It has been found that an increase in dextran molecular mass causes an increase in RBCs aggregation, and this may also be true for conglomerates of oxidized albumin such AOPPs and ALEs [100]. Furthermore, in one study an increase in the aggregation of RBCs and platelets was found to be associated with the degree of albumin oxidation [101]. There is a strong link between the aggregation of RBCs and thrombosis [102,103,104]. Thrombotic events are frequently observed in COVID-19 patients and are associated with increasing disease severity, as well as contributing significantly to death [105,106]. It has been found that AOPPs can also directly contribute to coagulation abnormalities by activating platelets via a CD36-mediated signaling pathway [32,33]. Many studies have revealed that serum albumin is inversely associated with artery and venous thrombosis events [107]. It has been hypothesized that albumin has anticoagulant and antiplatelet activities, probably due to its antioxidant effect [107]. Albumin increases fibrinolysis and inhibits erythrocyte aggregation. In addition, albumin neutralizes fibrinogen binding to endothelial cells, thus antagonizing several prothrombotic effects of fibrinogen [108]. Additionally, endothelial dysfunction and blood viscosity are increased in patients with hypoalbuminemia [109]. During inflammatory states, the coagulation cascade favors thrombus formation due to the decreased synthesis and increased catabolism of albumin. Hypoalbuminemia and hypercoagulability also coexist in patients with severe COVID-19 [103]. Our results revealed that D-dimers were moderate negatively correlated with albumin concentration (r = −0.477), positively associated with AOPPs level (r = 0.453), and positively associated with the AOPPs/albumin ratio (r = 0.534) in COVID-19-related pneumonia patients. D-dimers are products of the degradation or breakdown of fibrin, which are formed during blood clotting processes. This study provides the first evidence that oxidized albumin may be involved in hypercoagulability in patients infected with SARS-CoV-2.

The therapeutic effect of the drug is determined by its unbound fraction. In one of our studies, it was found that the oxidation of HSA by chloramine T reduced the binding activity of HSA [110]. Similar conclusions can be drawn from the studies of other researchers [111,112]. Some studies have also reported a reduction in the binding ability of HSA due to aldehyde modification [113]. Moreover, we observed that the free fraction of drugs increased in healthy patients depending on the amount of AOPPs present. Therefore, the pharmacokinetics of drugs used in the treatment of COVID-19 patients should be investigated. This study highlights the issues of drug treatment during COVID-19 infection, as medications may cause toxic effects. These issues seem to be very important for drugs that are closely bound to HSA and that have a narrow therapeutic index. Some observed abnormities in the levels of liver enzymes may be directly associated with drug-induced liver injury. Because the SARS-CoV-2 spike protein binds to albumin, this affinity may be influenced by oxidative stress, aldehyde modification, or the glycation of albumin. Indeed, it was recently observed that glycation reduced the binding ability of albumin [25]. These findings should be extended to the testing of virus binding to AOPPs or ALEs.

Infusions of human albumin solution can reduce the more severe or life-threatening events that take place in COVID-19 patients [38,114]. However, there is no consensus as to whether simply administering albumin to patients with hypoalbuminemia reduces their morbidity and mortality [48,115,116,117]. Hypoalbuminemia itself is not always considered a cause of the underlying pathology [48]. On the other hand, some studies have revealed that therapy with intravenous albumin may improve organ function, respiratory status, and ventilation-perfusion matching in critically ill patients with hypoalbuminemia or patients with acute respiratory distress syndrome [118,119]. One study revealed that albumin supplementation is able to reduce hypercoagulability in SARS-CoV-2, which was confirmed by the observation of a marked reduction in D-dimers [108]. Clinical trials and extensive studies on infusions of serum albumin in COVID-19 patients are urgently needed.

As albumin is taken up by endocytosis in infected cells in greater volumes in conditions of inflammation, the use of conjugate serum albumin with antiviral drugs to effectively target the extracellular and intracellular viral components is highly recommended in the treatment of patients infected with SARS-CoV-2.

The co-existence of hypoalbuminemia and oxidative stress in many diseases may lead to the hypothesis that oxidative modifications of albumin decrease its detection and influence albumin quantification. In our patient group, albumin was found to be weakly but significantly negatively correlated with AOPPs (r = −0.323). Some in vitro studies have shown the decreased detection of oxidized albumin by commonly used clinical assays [26,27,30,120]. This may explain the close association observed between a lower concentration of albumin and increased disease severity and mortality in patients with COVID-19 in many studies. This ‘‘apparent” hypoalbuminemia should be clarified and checked. Aldehydes and endogenous substances formed during COVID-19 infection by binding to albumin can also compete for binding sites with bromocresol green or purple, which are commonly used in the detection of albumin concentration. The effect of aldehyde interactions on albumin detection is currently unknown.

The strength of our study was to compare AOPPs levels for the same COVID-19 patients during and long after passing the acute stage of infection. Such a comparison allowed us to formulate the suggestion that oxidized albumin is closely linked to the pathophysiology of COVID-19 infection as the demographic characteristics and comorbid conditions remain similar. This study also has some limitations. The blood samples were taken from a single-center donation. Critically ill patients demanding invasive mechanical ventilation were excluded from this study. The sample size of examined patients was decreased in the following blood collections. Some patients recovered during the first week after being admitted and left the hospital before the second blood sampling. After 6- and 12-months post infection, some patients died, were hospitalized due to comorbidities, or were unwilling to continue participating in the study.

## 4. Materials and Methods

### 4.1. Clinical Study

In this study, we included patients hospitalized in the Department of Lung Diseases, Neoplasms, and Tuberculosis of the Regional Center of Pulmonology in Bydgoszcz, Poland, from April to December 2021. Patients who had been hospitalized with COVID-19 pneumonia that had been confirmed by a positive reverse-transcription polymerase-chain-reaction (RT-PCR) test result from a nasopharyngeal swab according to the World Health Organization (WHO) criteria [121] and radiographic imaging (HRCTs were performed using a 64-slice Siemens Somatom Sensation (Siemens Healthcare, Erlangen, Germany) system with a slice thickness ≤0.5 mm) or chest X-ray were eligible for enrolment. Patients had a blood oxygen saturation below 94% while breathing ambient air but were excluded if they were receiving continuous positive airway pressure, bilevel positive airway pressure, or mechanical ventilation. There were 13 women and 53 men among the subjects, and the mean age of all patients was 62.3 years. Most of the patients had a history of comorbidities. The most common were cardiovascular diseases (in 51% of patients), type 2 diabetes (in 21%), previous lung diseases (in 9%), and cancer (in 6%). All patients had undergone basic laboratory tests assessing the advancement of inflammation, the function of their liver and kidneys, and the parameters of their coagulation system. The mean percentage of lung involvement as assessed by the application of CT pneumonia analysis was 28%. The study group of COVID-19 patients was also tested one week after their admission to the hospital, as long as they were in the acute phase of infection. The group consisted of 11 women and 33 men, and the mean age of the patients was 62.7 years. The study group of COVID-19 patients was invited back for us to take another round of blood samples after 6 months. A total group of 30 people applied—5 women and 25 men—and the mean age of all patients was 60.6 years. After another 6 months, the AOPPs tests for 27 patients (3 women and 24 men, mean age 60.3 years) were repeated.

### 4.2. Sample Preparation

A 4 mL blood sample was taken for examination from each subject included in the study. For each COVID-19 patient, the sample was drawn on the same day or the next upon admission to the hospital department. All samples were processed within 2 h of collection. Blood samples were added to standard sterile polystyrene tubes containing EDTA and then centrifuged at 3500 rpm at 4 °C for 5 min to obtain plasma. The plasma fraction was collected and stored at −80 °C until measurement. Multiple freeze–thaw cycles were avoided. Measurements were taken within 1 h of defrosting the sample.

### 4.3. In Vitro Study

Chloramine T hydrate, sodium thiosulfate, malondialdehyde tetrabutylammonium salt (MDA), and HSA were received from Sigma-Aldrich. ALEs-HSA were obtained by the in vitro incubation of HSA (100 µM) with MDA (10 mM) at 37 °C. Solutions were suspended in PBS at pH = 7.4. AOPPs-HSA were produced in vitro by treating purified HSA (100 µM) with chloramine T at different concentrations from 0.37 to 4 mM in a volume of 1 mL. Chloramine T hydrate was added immediately after dissolution. The time of incubation was 60 min. Then, sodium thiosulfate (20 µL) at a concentration 4 times higher than that of chloramine T was added to eliminate the residue of unreacted chloramine T hydrate molecules. The experiments with chloramine T were repeated 3 times.

### 4.4. AOPPs Measurements

The level of AOPPs was determined by measuring the absorbance at 340 nm, according to the modified method described for the first time by Witko-Sarsat [51]. Briefly, the reactant mixture used for the AOPPs assay contained 1.875 mL of 0.2 M citric acid and 25 µL of 1.16 M potassium iodide. Then, 1.9 mL of this mixture was added to 100 µL of the test sample (plasma, HSA, modified has, or PBS-blank), and the absorbance was recorded immediately. In contrast with the original method, citric acid was used instead of acetic acid. This modified method is characterized by a greater stability over time [122]. The results were expressed as chloramine T equivalents.

### 4.5. Time-Resolved Fluorescence Spectroscopy Measurements

A time-resolved spectrofluorometer Life Spec II (Edinburgh Instruments Ltd., Livingston, United Kingdom) with a sub-nanosecond pulsed EPLED^®^ diode emitting light at a wavelength of 360 nm was used to measure the fluorescence lifetime of the plasma. Plasma samples were not diluted. The exposure time of the samples was 5 min. Fluorescence measurements of the plasma were conducted at wavelengths of 450 nm. The measurements were carried out with the use of quartz 3.5 × 10 mm cuvettes. The fluorescence lifetimes were obtained by the deconvolution analysis of the data using the multiexponential model of fluorescence decay, and the instrument response function was taken into account. Then, the mean fluorescence lifetime value was calculated as the weighted average of fluorescence lifetimes obtained from the three-exponential model of fluorescence decay. As averaging weights, the contributions of individual components (areas under decay curves) to the total fluorescence were calculated. The appropriate number of exponents was determined on the basis of Chi-square (χ^2^) statistical analysis and the visual assessment of residual plots.

### 4.6. Statistical Analysis

The preliminary step of the statistical analysis was the Shapiro–Wilk test of the normality of the distribution of the measured parameters. Due to the non-normality of the part-analyzed variables, the dependencies were determined by Spearman’s rank correlation coefficients (r values). The differences between the measurements of AOPPs recorded at different times (on admission, one week later, 6- and 12-months post infection) in the COVID-19 group were compared with the Wilcoxon signed-rank test and were considered significant at *p* < 0.05.

## 5. Conclusions

Serum albumin measurement may serve as a predictor of disease severity in patients with COVID-19. Our findings enhance the knowledge of the role of oxidized albumin in SARS-CoV-2 infection. The correlations observed between the level of albumin or AOPPs with inflammatory parameters or surrogate markers of COVID-19 severity as well as with lung HRCT score confirm the involvement of oxidative stress in COVID-19. Our in vitro study with chloramine T-induced albumin oxidation provides further insight into the mechanisms of the “cytokine storm” observed in COVID-19-related pneumonia. The modifications of albumin by ROS and aldehyde in the bloodstream and tissue cells outside the vascular bed caused by COVID-19 infection may have significant clinical implications and should be further explored at the earliest possible opportunity. The detection of “apparent” hypoalbuminemia should also be clarified.

## Figures and Tables

**Figure 1 ijms-23-10103-f001:**
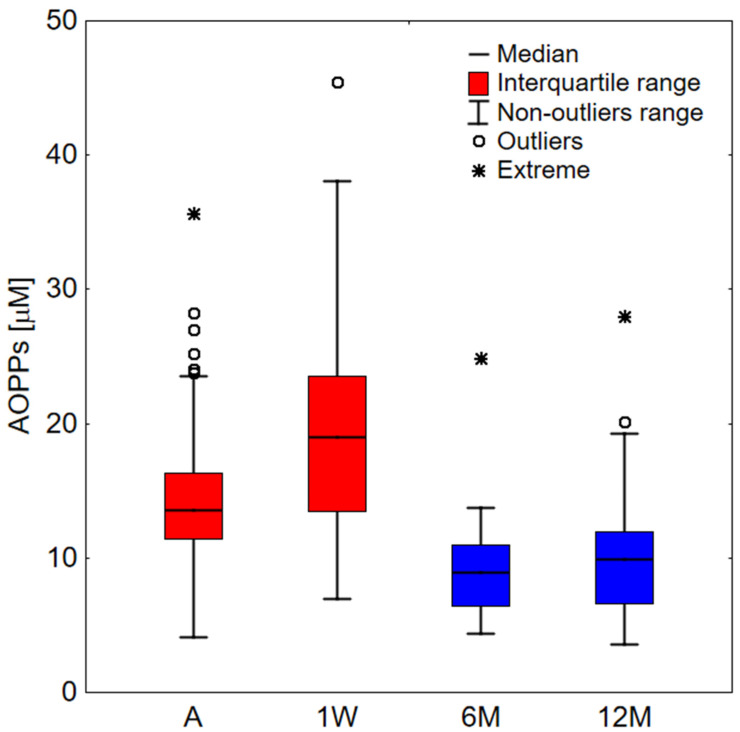
AOPPs values for a study group. A—COVID-19 patients on admission to hospital (*n* = 66); 1W—COVID-19 patients one week upon admission (*n* = 44); 6M—COVID-19 patients after 6 months from infection (*n* = 30); 12M—COVID-19 patients after 12 months from infection (*n* = 27). Data far away by more than 1.5 and 3 interquartile range from this range were considered outliers and extreme, respectively.

**Figure 2 ijms-23-10103-f002:**
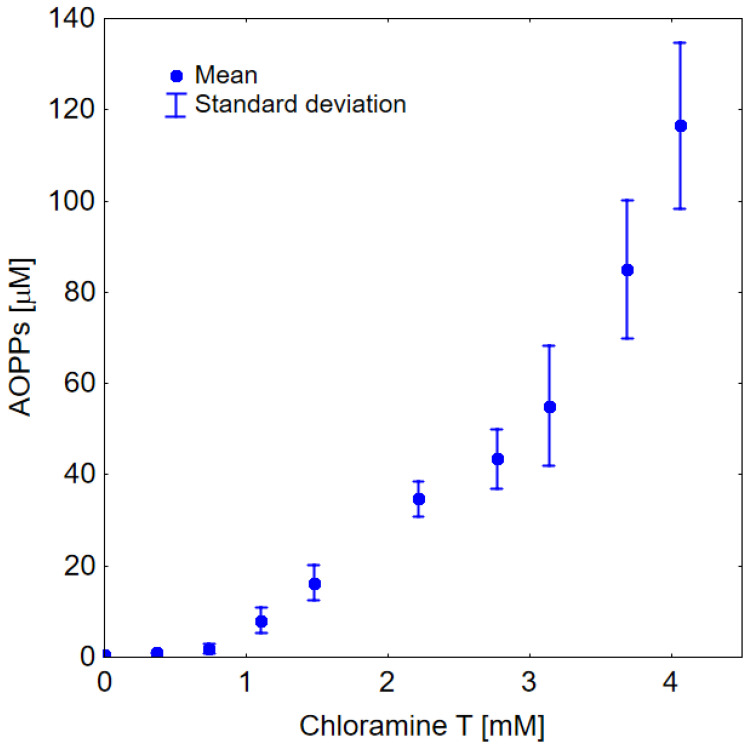
The formation of AOPPs dependence on chloramine T concentration added to HSA (100 µM). The measurements were repeated 3 times.

**Figure 3 ijms-23-10103-f003:**
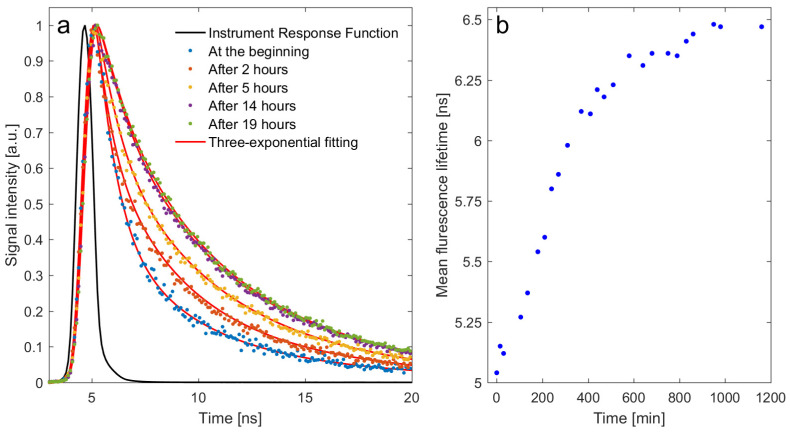
(**a**) The normalized fluorescence decay curves of ALEs-HSA at 450 nm at different times of incubation after the addition of MDA to HSA. (**b**) Mean fluorescence lifetime dependence on time of incubation HSA with MDA.

**Table 1 ijms-23-10103-t001:** Demographic, clinical, and laboratory parameters and their correlation with albumin, AOPPs, and the ratio of AOPPs/albumin of studied patients with COVID-19 pneumonia on admission. Red-colored values indicate statistical significance (*p* < 0.05). WBC, White blood cells count; RBC, Red blood cells count; Hgb, Hemoglobin; PLT, Platelets count; CRP, C-reactive protein; LDH, Lactate dehydrogenase; CPK, Creatinine phosphokinase; AST, Aspartate transaminase; ALT, Alanine transaminase; IL-6, Interleukin-6; HRCT, High-resolution computed tomography; AOPPs, Advanced oxidation protein products.

Parameters (Units)	Median	Interquartile Rang	Reference Values	Albumin		AOPPs		AOPPs/Albumin	
				r	*p*	r	*p*	r	*p*
Age (years)	64.5	51–72		** −0.260 **	** 0.035 **	−0.021	0.868	0.068	0.585
Symptoms (days)	7	5–10		−0.142	0.256	** 0.318 **	** 0.009 **	** 0.295 **	** 0.016 **
WBC (10^3^/µL)	6.8	5.1–9.6	4.0–10.0	−0.214	0.084	−0.035	0.778	0.055	0.661
Neutrophils (10^3^/µL)	4.8	3.6–7.5	2.5–5.0	−0.192	0.125	0.005	0.969	0.075	0.553
Lymphocytes (10^3^/µL)	0.9	0.7–1.2	1.5–3.5	−0.031	0.805	−0.115	0.363	−0.072	0.571
RBC (10^6^/µL)	4.45	4.2–4.8	4.5–5.5	0.194	0.118	−0.162	0.195	−0.201	0.106
Hgb (g/dL)	13.7	12.7–14.5	14.0–18.0	0.113	0.365	−0.172	0.168	−0.181	0.146
PLT (10^3^/µL)	211	172–292	130–350	−0.163	0.192	−0.012	0.923	0.048	0.702
CRP (mg/L)	84	42–138	<5.0	−0.235	0.058	** 0.323 **	** 0.008 **	** 0.357 **	** 0.003 **
Procalcitonin (ng/mL)	0.08	0.05–0.15	<0.05	** −0.480 **	** <0.001 **	** 0.309 **	** 0.012 **	** 0.388 **	** 0.001 **
LDH (U/L)	645	543–885	225–450	** −0.437 **	** <0.001 **	** 0.362 **	** 0.003 **	** 0.459 **	** <0.001 **
D-Dimers (ng/mL)	941	735–1574	<500	** −0.477 **	** <0.001 **	** 0.453 **	** <0.001 **	** 0.534 **	** <0.001 **
Troponin (ng/L)	10.6	6.2–21	<19.0	** −0.268 **	** 0.030 **	0.238	0.054	** 0.309 **	** 0.012 **
Creatinine (mg/dL)	0.97	0.87–1.14	0.8–1.3	−0.067	0.591	0.162	0.195	0.177	0.155
CPK (U/L)	142	76–231	25–200	0.008	0.952	0.147	0.238	0.120	0.336
AST (U/L)	53	36–70	<37	** −0.307 **	** 0.012 **	** 0.362 **	** 0.003 **	** 0.417 **	** <0.001 **
ALT (U/L)	43	30–66	<40	−0.193	0.120	** 0.245 **	** 0.047 **	** 0.282 **	** 0.022 **
IL-6 (pg/mL)	13.1	4.8–36	<7.0	−0.009	0.942	−0.171	0.177	−0.166	0.189
HRCT score	0.25	0.14–0.41		** −0.423 **	** <0.001 **	** 0.348 **	** 0.004 **	** 0.433 **	** <0.001 **
Albumin (g/L)	34	31–37	39–51	1.000	-	** −0.323 **	** 0.008 **	** −0.593 **	** <0.001 **
AOPPs (µM)	13.5	11.4–16.3		** −0.323 **	** 0.008 **	1.000	-	** 0.932 **	** <0.001 **
AOPPs/Albumin (µM/g)	0.39	0.33–0.50		** −0.593 **	** <0.001 **	** 0.932 **	** <0.001 **	1.000	-

**Table 2 ijms-23-10103-t002:** The level of significance (*p*-values) between the study group. A—COVID-19 patients on admission to hospital; 1W—COVID-19 patients one week upon admission; 6M—COVID-19 patients after 6 months from infection; 12M—COVID-19 patients after 12 months from infection. Red-colored values indicate statistical significance (*p* < 0.05). For data comparison, we used AOPPs measurements from the same patients but recorded at different times.

	A	1W	6M	12M
A		** <0.001 **	** <0.001 **	** 0.016 **
1W	** <0.001 **		** <0.001 **	** 0.004 **
6M	** <0.001 **	** <0.001 **		0.055
12M	** 0.016 **	** 0.004 **	0.055	

## Data Availability

The data presented in this study are available on request from the corresponding author.

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
