# Peer review of "Study of Albumin Oxidation in COVID-19 Pneumonia Patients: Possible Mechanisms and Consequences"

_ijms, 2022, doi:10.3390/ijms231710103_

Round 1

Reviewer 1 Report

This is a very meaningful and interesting work, thus I would like to recommend it for publication after minor revision:

1.     The presentation mode of this article is not the typical format of IJMS. Please check the template and revise it carefully.

2.     Page 5, in Section 2.6. Statistical analysis, the authors stated that "The differences were considered significant at p <0.05.".  However, where are the p values in the manuscript, and please mark which samples were compared.

3.     Figure 1, “mean ± SD, and n=?” should be added to the legend.

4.     Figure 2, “n=?” should be added to the legend..

5.     As an article, more data should be added to the manuscript.

6.     The dread of covid-19 is that it will affect other inflammation related complications or basic diseases, such as intestinal diseases <engineered regeneration 2 (2022) 279-287. > If possible, the author can introduce it in a short sentence in the introduction

Author Response

Answer to Reviewer 1

Thank you for your earlier comments

  1. The presentation mode of this article is not the typical format of IJMS. Please check the template and revise it carefully.

Answer

       It was encouraged to use the journal template to prepare the manuscript but it was not obligatory. The editor changes it.

  1. Page 5, in Section 2.6. Statistical analysis, the authors stated that "The differences were considered significant at p <0.05.".  However, where are the p values in the manuscript, and please mark which samples were compared.

Answer

     We clarified the sentences and caption of table and figure in which we mentioned the p values and which data we compared. The significant correlation (p <0.05) of albumin, AOPPs, and the ratio of AOPPs/albumin with clinical assays in patients with COVID-19 pneumonia on admission was marked by red-coloured in Table 1. The differences between the measurements of AOPPs recorded at different times (on admission, one week later, 6 and 12 months post infection) in the COVID-19 group were compared with Wilcoxon signed-rank test and were considered significant at p <0.05. The level of significance (p-values) between the study group was shown in Table 2. Red-coloured values indicate statistical significance (p < 0.05). For data comparison, we used AOPPs measurements from the same patients but recorded at different times. P values in the manuscript were placed in the chapter results.

  1. Figure 1, “mean ±SD, and n=?” should be added to the legend.

Answer

As was stated in the materials and methods the preliminary step of the statistical analysis was the normality test of the distribution of the measured parameters. Due to the non-normality of the analyzed variables, instead of the mean and standard deviation, the median and interquartile range should be used to present the obtained results just like we used the non-parametric tests instead of parametric ones to compare them. On the other hand, introducing the mean and standard deviation together with the median and interquartile range in Figure 1 will make it unreadable. We added the number of patients to the caption of Figure 1.

  1. Figure 2, “n=?” should be added to the legend.

Answer

        We added the number of measurements to the caption of Figure 2

  1. As an article, more data should be added to the manuscript.

Answer

     Demographic, clinical, and laboratory parameters and their correlation with albumin, AOPPs, and the ratio of AOPPs/albumin of studied patients with COVID-19 pneumonia on admission. was also shown in Table 1. In figure 2 it was shown changes in AOPPs levels recorded during the different times (on admission, one week later, 6 and 12 months post infection) in the COVID-19 group. In our opinion, the amount of data is sufficient, but we have corrected the captions for tables and figures.

  1. The dread of covid-19 is that it will affect other inflammation related complications or basic diseases, such as intestinal diseases <engineered regeneration 2 (2022) 279-287. > If possible, the author can introduce it in a short sentence in the introduction

Answer

We added this reference to a created sentence in the introduction

Reviewer 2 Report

Th paper is interesting and it sounds that can contribute to the field with valuable information. However the manuscript needs some modifications and the results presentation have to be redesigned. The references beside covid have to be updated.

Congratulation for your work and efforts to prepare the manuscript. Without any doubt Covid-19 pneumonia is nowadays considered as a polysystemic syndrome while all pathophysiological processes leading to various tissues derangements are consequences of the inflammation storm. Your research is adding valuable information in the field. However I have some concerns which are the following.

Comment 1

In the abstract (2nd and 3rdline) is stated that "transport of albumin .....is exaggerated". To my knowledge albumin does not really enters a cell but is taken up by endocytosis into the lysosomal compartment. I advise you either to be more specific or to rephrase the sentence.

Comment 2

In the introduction chapter the hypothesis of the study as well as the aims are not clearly defined.

Comment 3

In the Material and Methods chapter are referred 2 distinct number of participants (13 -53 & 11-33).

Comment 4

In the results chapter as you state, the control subjects had to be chosen not only age-gender-weight matched but for the presence of comorbidities similar to study population. Otherwise any statistical evaluation will have weakened significance.

Comment 5

Please include study limitation and strengths in the Discussion chapter.

Comment 6

Please update your non covid references.

Author Response

Answer to Reviewer 2

Thank you for your earlier comments

Comments and Suggestions for Authors

The paper is interesting and it sounds that can contribute to the field with valuable information. However the manuscript needs some modifications and the results presentation have to be redesigned. The references beside covid have to be updated.

Congratulation for your work and efforts to prepare the manuscript. Without any doubt Covid-19 pneumonia is nowadays considered as a polysystemic syndrome while all pathophysiological processes leading to various tissues derangements are consequences of the inflammation storm. Your research is adding valuable information in the field. However I have some concerns which are the following.

Answer

Thank you very much for those words. We appreciate it very much.

Comment 1

In the abstract (2nd and 3rdline) is stated that "transport of albumin .....is exaggerated". To my knowledge albumin does not really enters a cell but is taken up by endocytosis into the lysosomal compartment. I advise you either to be more specific or to rephrase the sentence.

Answer

We looked at the literature and other studies and realized that this process has not been yet entirely elucidated, especially in infectious diseases, including viral ones. We decided to delete this statement in order not to confuse the readers.

Comment 2

In the introduction chapter the hypothesis of the study as well as the aims are not clearly defined.

Answer

We added the hypothesis of the study as well as the aims in the introduction chapter

Comment 3

In the Material and Methods chapter are referred 2 distinct number of participants (13 -53 & 11-33).

Answer

The second numbers (11-33) were for patients examined a week later after admission as long as they were in the acute phase of infection and had not left the hospital. We added following sentences at the end of the discussion in the study limitations: “ The sample size of examined patients was decreased in the following blood collections. Some patients recovered during the first week after being admitted and left the hospital before the second blood sampling. After 6 and 12 months post infection some patients died, were hospitalized due to comorbidities or were unwilling to continue participating in the study.   

Comment 4

In the results chapter as you state, the control subjects had to be chosen not only age-gender-weight matched but for the presence of comorbidities similar to study population. Otherwise any statistical evaluation will have weakened significance.

Answer

We also agree that any statistical evaluation will have weakened significance when COVID-19 patients are compared to healthy without comorbidity and younger. Therefore, after deep reflection, we decided to remove the healthy group from the study. In our opinion, the best comparison is to compare the same patients during the COVID-19 infection and a few months post-infection. We realized that this was in fact the strength of our study. The demographic characteristics and comorbid conditions of patients remain similar. The aim of the study was to check if oxidized albumin (AOPPs) is elevated during COVID-19 infection. So, thanks to this comparison, we can prove easily our hypothesis.

Comment 5

Please include study limitation and strengths in the Discussion chapter.

Answer

We added study limitations and strengths at the end of the Discussion chapter.

Comment 6

Please update your non covid references.

Answer

We tried to update the literature. However, we could not change a few of them, because in our opinion they are much more important than newer reports or the knowledge about a given topic has not changed or there was no new research.
